# Phosphorus/Bromine Synergism Improved the Flame Retardancy of Polyethylene Terephthalate Foams

**DOI:** 10.3390/polym16121690

**Published:** 2024-06-13

**Authors:** Jia Du, Jiaxin Zheng, Chunling Xin, Yadong He

**Affiliations:** College of Mechanical and Electronically Engineering, Beijing University of Chemical Technology, Beijing 100029, China17801190249@163.com (J.Z.); heyd@mail.buct.edu.cn (Y.H.)

**Keywords:** polyethylene terephthalate, flame retardant, foamability, limiting oxygen index, char layer

## Abstract

Polyethylene terephthalate (PET) foams have the characteristics of being lightweight and high strength, as well as offering good heat resistance, minimal water absorption, etc., and they have been widely used in the wind power field. In addition, they are being promisingly applied in automotive, rail, marine, construction, and other related fields. Therefore, the flame retardancy(FR) of PET foams is an issue that requires investigation. The addition of flame retardants would affect the chain extension reaction, viscoelasticity, and foamability of PET. In this study, zinc diethyl hypophosphite (ZDP) and decabromodiphenylethane (DBDPE) were used to form a synergistic FR system, in which ZDP is an acid source and DBDPE is a gas source, and both of them synergistically produced an expanded carbon layer to improve the flame retardancy of PET foams. The ratio of ZDP and DBDPE is crucial for the carbon yield and the expansion and thermal stability of the char layers. At the ZDP/DBDPE ratios of 9/3 and 7/5, the thickness of the char layers is about 3–4 mm, the limiting oxygen index (LOI) values of FR modified PET are 32.7% and 33.6%, respectively, and the vertical combustion tests both reached the V-0 level. As for the extruded phosphorous/bromine synergism FR PET foams, ZDP/DBDPE ratios of 3:1 and 2:1 were applied. As a result, the vertical combustion grade of foamed specimens could still reach V-0 grade, and the LOI values are all over 27%, reaching the refractory grade.

## 1. Introduction

Polyethylene terephthalate (PET) foam is characterized by lightweight, high strength, and good thermal resistance qualities [1,2]. It is one of the structural core materials which have been widely used in the wind power field [3]. In addition, PET foam has low thermal conductivity, low water absorption, no rot, no mildew, easy to process and other characteristics, so it also would have a wide range of applications in automobiles, rail transportation, construction, ships, and other prospects [4,5,6]. Therefore, the flame retardancy of PET foams has become more and more critical, which affects its wide application in construction, automobiles and rail, etc. PET is a flammable material and is prone to dripping due to the random chain-breaking mechanism during combustion [7,8], and the limiting oxygen index is only about 21%. The flame retardancy of PET foams is further reduced due to the increased contact area with air [7,8,9]. In order to improve the flame-retardant properties of PET and its foams to make them suitable for more applications, meet national safety standards, and reduce fire hazards, adding flame retardants is a common method to reduce the flammability of PET product.

Before flame-retardant modification of PET, it is necessary to understand its combustion mechanism. The polymer combustion process has five stages: thermal decomposition, ignition, combustion propagation and development, fully stabilized combustion, and combustion decay. The combustion region can be divided into three levels: the polymer surface, condensed phase, and gas phase [10]. The thermal decomposition temperature of PET in air is 283–306 °C, and the thermal decomposition temperature in nitrogen can reach 420 °C [11]. At this stage, the PET molecular chain begins to undergo random cleavage, and the relative molecular mass significantly decreases. As the thermal decomposition continues, it begins to produce flammable gases, and in the presence of an ignition source and oxygen, the PET will be ignited. After the ignition, the solid and gas phases will undergo exothermic reactions, releasing a large amount of heat. The heat is transferred to the surface of the PET matrix or even inside and spreads to the surrounding areas, continuing to promote the cracking of PET molecular chains to produce flammable gases; flammable gases escape to the flame zone and are ignited after contacting the air at high temperatures, which makes the flame begin to stabilize the combustion when it reaches the critical point. At the end of PET matrix degradation, the flammable gas generated, and which escapes to the flame area, decreases, and the heat generated by combustion also decreases. At this time, the combustion can no longer be maintained and, thus. attenuated until extinguished. The combustion of the products of PET cracking follows a chain reaction [12], and to slow or stop the combustion, it is necessary to stop the chain reaction and reduce the heat release.

Flame retardants are characterized by their flame-retardant elements [13,14]. For example, the halogen elements fluorine, chlorine, bromine, iodine, etc., are very effective combustion inhibitors of hydrocarbons because they capture the free radicals in the gas phase and, thus, interrupt the chain reaction [15]. Common halogenated flame retardants include decabromodiphenyl ether [16], chlorinated paraffin [17], etc. Due to their highly hazardous effects on the environment and the human body, halogenated flame retardants are constantly being restricted, and the research related to flame retardancy is gradually shifting to non-halogenated flame retardants [18,19]. Halogen-free flame-retardant elements include phosphorus, nitrogen, boron, silicon, ammonia, aluminum, magnesium, antimony, etc. [20], and the flame retardants containing these elements play a flame-retardant role mainly in the gas phase or condensed phase. Common halogen-free flame retardants include red phosphorus [21], phosphonitrile [22,23], phosphonate [24,25], polyphosphoric acid [26], zinc borate [27], magnesium hydroxide [28], aluminum hydroxide [29], antimony trioxide [30], etc.

In the flame-retardant(FR) modification of PET, a single FR element usually cannot meet the requirements for the FR properties of PET, so it is necessary to select a suitable FR system to jointly play a FR role to achieve a good FR effect. Ordinary FR systems include halogen/antimony synergism [31], phosphorus/halogen synergism [32,33], phosphorus/nitrogen synergism [34], phosphorus/phosphorus synergism [35,36], and nitrogen/phosphorus/silicon synergism [37], etc. The mechanism of these synergistic systems can be broadly categorized into two types: one is mainly flame retardant in the condensed-phase; the other type mainly plays a flame retardant role in the gas-phase.

PETproduces dripping during combustion [7,8]. Although the dripping can take away a large amount of heat to slow the matrix combustion down or stop it, the burning droplets will likely ignite other combustibles, leading to the spread of fire. Given the characteristics of PET, which often drips during combustion, many people try to add co-efficacies in phosphorus-containing flame retardants or co-polyesters to achieve the anti-dripping effect. Zhao et al. [38] synthesized phosphorus-containing and ionomer monomers and phosphorus-containing and high-temperature self-cross-linking monomers, and copolymerized the two monomers into PET molecular chains to obtain anti-dripping PET co-polyesters, respectively. Among them, the ionomer works through the ionic aggregation effect and, thus, forms a reversible physical cross-linking network in the matrix to improve the viscosity of the melt during combustion and inhibits the dripping. At the same time, the high-temperature self-cross-linking has a better effect, which is anti-dripping through the formation of the irreversible cross-linking network by a chemical reaction at high temperatures. Furthermore, Guo et al. [39] successfully addressed the issue of dripping by incorporating a novel arylene ether-containing monomer (PBPBD) into the PET main chain under elevated temperatures; the PBPBD structural units undergo rearrangement reactions to form a conjugated heterocyclic aromatic structure, which promotes the formation of a char layer during combustion of the PET, and at the same time effectively enhances the viscosity of the PET.

It can be found that the flame-retardant modification of PET generally selects the synergistic system that mainly works in the condensed phase and that also actsin the gas phase, such as phosphorus/halogen synergistic systems, phosphorus/nitrogen synergistic systems, etc. Green [40] found that the flame-retardant efficiency of phosphorus is much higher than bromine when using brominated phosphonate for flame-retardant PET/PC mixtures. This is further improved when the two are synergistically flame-retardant.

In this study, a phosphorus/bromine synergistic system for flame-retardant modification of PET was used; ZDP has high phosphorus content and a flame-retardant effect in both the condensed and gas phases [41]. There is no chemical bonding between DBDPE and PET. DBDPE has the advantages of high thermal stability, bromine content, and flame-retardant efficiency [42], and is a typical representative of additive flame retardants. In this paper, ZDP and DBDPE were used to form a phosphorus–bromine synergistic system to prepare flame-retardant PET, in which ZDP was used as an acid source, and DBDPE was used as a gas source to study the effect of the two complexes on the flame-retardant and foaming properties of PET.

## 2. Materials and Methods

### 2.1. Materials

The PET pellets used in this study were provided by Sinopec Yizheng Chemical Fiber Co., Ltd. (Yizheng, China). Their intrinsic viscosity was 0.8 dL/g (determined in a phenol and tetrachloroethane mixture with a phenol to tetrachloroethane ratio of 3:2); the glass transition temperature was about 70 °C, and the melting temperature was about 250 °C. ZDP (99.75% purity), used as a flame retardant, was bought from Clariant Chemicals Ltd. (Shanghai, China). Its phosphorus mass fraction was 20.0 wt%. The DBDPE (99.97% purity) was purchased from the Shandong Shouguang Weidong Chemical Co., Ltd. (Shouguang, China), model RDT-3, with a melting point greater than or equal to 340 °C and a bromine content equal to 81.57%. Pyromellitic dianhydride (PMDA) was supplied by the Aladdin Reagent (Shanghai) Co., Ltd. (Shanghai, China). The talc content of the PET-based nucleating agent masterbatch was 20 wt%, which was homemade in the laboratory. Diiodomethane purchased from the Aladdin Reagent (Shanghai) Co., Ltd.; the purity was 98%, and it contained stabilizing copper chips.

### 2.2. Preparation of Flame-Retardant PET Composites

#### 2.2.1. Reactive Blending

PET, ZDP, DBDPE, and PMDA were dried under vacuum conditions for 12 h with the temperature set at 120 °C. The dried materials were weighed according to the ratios in Table 1 and added to the Haake Polylab OS. The temperature of the Polylab was set at 275 °C, and the rotor speed was set to 60 rpm. The obtained blended samples were used for subsequent thermogravimetric analysis, rheological tests, and batch foaming.

#### 2.2.2. Flame-Retardant Sample Preparation

The dried materials were weighed according to the ratios in Table 1 and put into a high-speed mixer for premixing. The premixed material was added into the twin-screw extruder (ZSK25-WLE type), manufactured by W&P Company (Stuttgart, Germany), for extrusion cooling granulation. The feeding was set at 8 kg/h, the screw speed was 180 rpm, and the temperatures of each heating section of the extruder were set at 200 °C, 250 °C, 255 °C, 260 °C, 265 °C, 265 °C, 265 °C, 265 °C, 265 °C, 265 °C, and 260 °C, respectively. The extruded mixture was then cooled and pelletized. The extruded mixture of granules was first dried in an air blower drying oven at 100 °C for six h and then placed in a vacuum oven at 120 °C for 12 h, and then taken out and molded into standard flame-retardant samples using an injection molding machine (HTF120X2 type), provided by Haitian Group Co., Ltd. (Ningbo, China) with the temperatures of each section of the injection molding machine set to 240 °C, 250 °C, 265 °C, 265 °C, and 260 °C, respectively.

#### 2.2.3. Batch Foaming Experiment

The flame-retardant samples were dried under vacuum for 12 h in an oven, and the temperature was set at 120 °C. The autoclave (self-made) was heated up to 270 °C, the dried samples were put into it, and CO_2_ was injected to increase the pressure to 15 MPa and maintain it; the temperature was kept for 20 min and then cooled down to foaming temperature; after the temperature was stabilized, it was maintained for another 20 min, and then the pressure was relieved quickly, and the foamed samples were obtained.

#### 2.2.4. Flame-Retardant (FR) Masterbatch Preparation

ZDP and DBDPE were premixed in the ratio of ZDP:DBDPE = 2 or 3, then melt blended, extruded, cooled, and pelletized with PET in a twin-screw extruder at a temperature of 275 °C and a rotational speed of 300 rpm. The total content of ZDP and DBDPE in the flame-retardant masterbatch was 50 wt%.

#### 2.2.5. Flame-Retardant Foam Sample Preparation

The flame-retardant masterbatch, PET, nucleating agent masterbatch, and chain ex-tender were premixed according to the proportions in Table 1 and added to a tandem extruder foaming system. The temperature of the twin-screw extruder was set at 280 °C and the screw speed was set at 80 rpm; the temperature of the cooling single extruder was set at 235 °C, and the screw speed was set at 6 rpm. CO_2_ was used as the foaming agent.

### 2.3. Testing and Characterization

#### 2.3.1. Dynamic Rheological Testing

The Haake Mars III rheometer (Thermo Fisher Scientific, Waltham, MA, USA) was used to measure shear rheological properties. The samples, which were dried at 120 °C in a vacuum oven for 12 h, were placed on a steel parallel plate that was heated to 270 °C. The samples were then pressed into a 1 mm thickness in an inert environment, and the frequency sweeps were changed from 100 to 0.1 rad/s after starting. Meanwhile, to prevent oxidative degradation and hydrolysis caused by the contact of PET with air, N_2_ was passed through to protect it during the test.

#### 2.3.2. Thermogravimetric Analysis

Samples weighing about 10 mg were placed in an alumina ceramic crucible. A thermogravimetric analyzer (ZCT-B type), manufactured by Beijing Jingyi Hi-Tech Instrument Co., Ltd. (Beijing, China), recorded the quantity variation as the temperature rose from 100 °C to 800 °C in a nitrogen environment or air conditions.

#### 2.3.3. Vertical Combustion Test

Following test standard GB/T 2408-2021, horizontal and vertical combustion apparatus were used for vertical combustion experiments on the sample strips, with sample strips sized 125 mm × 13 mm × 3.2 mm. Ten seconds after the first ignition, we removed the ignition source and started measuring the time; the time when the flame was extinguished was recorded as t_1_; again, after ignition lasting for ten seconds, we removed the source of the ignition and started the time again; the time when the second flame was extinguished was recorded as t_2_, and we observed whether the dripping ignited the degreasing cotton placed under the sample.

#### 2.3.4. Limiting Oxygen Index Test

Following test standard GB/T2406.2-2009, at least 15 standard sample strips, with sizes of 80 × 10 × 4 mm, were prepared. We drew the marking line at 50 mm from the top of the sample strips for testing the lowest oxygen concentration to maintain the stable combustion of the sample strips. The conditions for determining the stable combustion of the sample were as follows: the flame crossed the marking line during the test, or the combustion time exceeded 5 min.

#### 2.3.5. Cone Calorimeter Test

Following test standard ISO-5660, the sample size was 100 mm × 100 mm × 50 mm; the sample was wrapped with aluminum paper to expose only the upper surface, radiant power of 50 Kw was employed, and the experimental process involved observing the changes in combustion.

#### 2.3.6. Scanning Electron Microscope

The surface of the char layer was treated with gold spraying in sections and then fixed on the sample stage for observation. Foam samples were cut with a thin blade, and then the sections were gold-sprayed for observation. The samples were observed under a scanning electron microscope (TM4000), which was manufactured by Hitachi, Japan. The cell dimensions and cell densities were measured and calculated using Image J (version 1.41).

#### 2.3.7. Determination of Foaming Ratio

We used the sponge densitometer (PMMD-A, Guance Test Instrument Co., Ltd., Beijing, China) to test the density of the sample before and after foaming, according to the Formula (1), to calculate the foaming ratio as follows:(1)$$\varphi =\frac{\rho_{1}}{\rho_{2}}$$
where *φ* is the foaming ratio, *ρ*_1_ is the density of the sample before foaming, and *ρ*_2_ is the density of the sample after foaming.

#### 2.3.8. Contact Angle Test

The samples were pressed into thin slices with a smooth surface. The sheet was placed on an operating platform and drops of water and diiodomethane droplets were then applied to the surface using a syringe purchased from Anhui Kangtai Medical Equipment Co., Ltd. (Hefei, China). The software measured the contact angle between the droplets and the sample surface, and the surface energy was calculated using the Owens–Wendt formula [43].

## 3. Results and Discussion

### 3.1. Flame Retardant Performance Analysis

#### 3.1.1. Vertical Combustion and the Limiting Oxygen Index Test

The vertical combustion rating and limiting oxygen index of the flame-retardant PETs after the addition of 12 wt% ZDP and DBDPE were tested, and the flame-retardant efficiency and synergistic effect parameters were calculated according to Equations (2) and (3); the results are shown in Table 2 and Figure 1. Compared with pure PET, the flame retardancy of modified PETs with the addition of 12 wt% ZDP or 12 wt% DBDPE was greatly improved. The limiting oxygen indexes increased from 21.6% to 29.8% and 28.7%, respectively; the burning time of the vertical combustion was also reduced to less than 10 s, but the UL-94 levels only reached V-2 due to the retardant’s inability to effectively inhibit the molten dripping.

The flame-retardant efficiency (*n*) was calculated according to the following Equation (2):(2)$$n=({LOI}_{PET/ZDP/DBDPE}-{LOI}_{pure\;PET})/(C_{P}+C_{Br})$$

The synergy efficiency (*E_Z/D_*) of ZDP and DEDPE was calculated according to the following Equation (3):(3)$$E_{Z/D}=\frac{\left({LOI}_{PET/ZDP/DBDPE}-{LOI}_{PET}\right)}{W_{ZDP}\left({LOI}_{PET/12ZDP}-I_{LO,PET}\right)+W_{DBDPE}({LOI}_{PET/12DBDPE}-{LOI}_{PET})}-1$$

For ZDP and DBDPE synergism retardants, the LOI of the PET/11ZDP/1DBDPE sample with 1 wt% DBDPE added increased to 31.2%, which is 1.4% higher compared to PET/12ZDP; at the same time; the dripping phenomenon disappeared, and the vertical combustion grade reached V-0. With the increase in DBDPE content, the LOI tended to go first up and then down. It reached a maximum value of 33.6% for the PET/7ZDP/5DBDPE samples, which was an improvement of 3.8% and 4.9% compared with the PET/12ZDP or PET/12DBDPE samples, respectively. The dripping phenomenon was suppressed at DBDPE contents higher than 4 wt% of total materials, and the vertical combustion grade was at the V-0 level.

From Table 2 and Figure 1, it can be seen that the flame-retardant efficiency n of PET/12DBDPE is only 0.7, while that of PET/12ZDP is 3.4. This indicates that the phosphorus element has higher efficiency for PET than the bromine element. With the increase in ZDP content, the *n* of ZDP/DBDPE blends increases continuously. As can be seen from Figure 1, the synergistic effect parameter *E_Z/D_* also showed a trend of increasing and then decreasing with ZDP content, which varied in the same way as the limiting oxygen index, with a maximum value for PET/7ZDP/5DBDPE. *E_Z/D_* was more significant than 0 for all PET/ZDP/DBDPE samples, especially at PET/7ZDP/5DBDPE, where *E_Z/D_* was 0.55. This indicates that ZDP and DBDPE have a synergistic effect in any ratio of ZDP/ DBDPE, while there is an optimum ratio with the best synergistic effect.

In order to verify the effect of the addition ratio of ZDP and DBDPE on the flame retardancy of PET/ZDP/DBDPE blends, the LOI values at ratios of ZDP/DBDPE = 2 and 3 with different total additions of ZDP/DBDPE were tested, and the results are shown in Figure 2. The flame retardancy deteriorated as the total additions reduced; however, the limiting oxygen indexes of the samples with a ratio of ZDP/DBDPE of 2 were higher than those of the samples with a ratio of 3 at any total additions. In particular, the LOI of ZDP/DBDPE with a ratio of 2 and a total additive amount of 8 wt% was even higher than that of ratio 3 and a total additive amount of 10 wt%. This confirms that the ratio of ZDP and DBDPE is crucial for flame synergistic of PET.

#### 3.1.2. Cone Calorimeter Test

The PET/ZDP/DBDPE mixture was subjected to a cone calorimeter test, which mainly evaluates the heat release, smoke, and poisonous gas, and the detailed data are shown in Table 3. The time to ignition (TTI) of the PET/12ZDP and PET/12DBDPE samples were both 40 s, and it can be found that the TTI of the PET/ZDP/ DBDPE samples increased, which indicates that the compounding of ZDP and DBDPE can reduce the flammability of PET, thus reducing the possibility of fire. Effective heat of combustion (EHC) reflects the gas-phase flame-retardant effect [44]. The EHC of PET/12ZDP was 27 MJ/kg, and it was generally reduced after adding DBDPE. This is because DBDBE, as a brominated flame-retardant, mainly acts as a gas-phase flame-retardant with free radical trapping as the primary role.

Heat release is mainly characterized by the heat release rate (HRR) and total heat release (THR), and its relationship with time is shown in Figure 3 and Figure 4. All samples reached the peak of HRR rapidly after ignition and then began to decrease. A small peak appeared around 240 s, after which it continued to decrease. This is the obvious feature of thin charred samples [45]. PET samples first burn rapidly after ignition. At the same time, a char layer will be formed on the burning surface under the action of the flame-retardant, which will inhibit the heat and mass transfer process during combustion, thus slowing down the thermal decomposition of the PET matrix. During continued combustion, the char layer is destroyed, and this results in previously isolated flammable gases entering the flame zone and reacting with the O_2_. As such, the second peak appears in the curve of heat release rate. The peak heat release rate PkHRR has been listed in Table 3 and shows the trend of first decrease and then increase when the ZDP content decreases. The minimum value of 437 kW/m^2^ was obtained at PET/7ZDP/5DBDPE, which is 15% lower than that of the PET/12ZDP sample. Meanwhile, the total heat release has the same trend, with 111 MJ/m^2^ and 105 MJ/m^2^ for PET/12ZDP and PET/12DBDPE samples, respectively. The minimum value of 84 MJ/m^2^ is for PET/7ZDP/5DBDPE. All these data show that the ZDP and DBDPE in the proper ratio form a practical synergistic effect, thus reducing the harm caused by the heat release when PET is burned.

Smoke and toxic gases are non-negligible lethal factors in fires [46]; this paper used total smoke production (TSP) and CO yields to evaluate the effect of ZDP and DBDPE compounding on the smoke and toxic gases produced during PET combustion. The TSP values of PET/12ZDP and PET/12DBDPE samples were 22 m^2^ and 24 m^2^, respectively. After the compounding of ZDP and DBDPE, the TSP was reduced to a minimum of 19 m^2^ for PET/9ZDP/3DBDPE samples, which indicates that the compounding of ZDP and DBDPE is effective in reducing flue gas generation. However, from the data on the CO production rate in Table 3, it can be seen that it increased significantly after the two compoundings, indicating the increase in toxic gas production.

#### 3.1.3. Carbon Residue Analysis

Figure 5 shows the residual char layer after the PET/ZDP/DBDPE blends were subjected to cone calorimeter testing. The residual carbon layers of the PET/12ZDP and PET/12DBDPE samples show completely different morphologies. The PET/12ZDP sample has a more extensive residual carbon layer, which can cover the surface of the tin foil. At the same time, there is almost no residual carbon layer of PET/12DBDPE after burning. This is related to the main flame-retardant mechanism of the samples; ZDP is an organophosphorus flame retardant, the decomposition of which produces H_3_PO_4_, HPO_3_, etc., which can promote the dehydration of PET to form a carbon layer and which can cover the carbon layer on the surface of the substrate to isolate the process of heat and mass transfer. In contrast, DBDPE is the bromine flame retardant, and its flame-retardant mechanism is to produce Br· and HBr free radicals after decomposition and to capture the free H· and OH· in the gas phase to interrupt the combustion reaction, so it does not help the charring of PET.

The thickness of the char layers for PET/9ZDP/3DBDPE and PET/7ZDP/5DBDPE increased to 3–4 cm and was much greater than that of the PET/12ZDP at 2 mm. This indicates that ZDP and DBDPE exhibit an intumescent flame-retardant mechanism. With ZDP as the acid source, DBDPE as the gas source, and PET as the char source, the expansion and flame-retardant mechanism works better under the appropriate ratio of ZDP and DBDPE, significantly increasing the thickness of the char layer. The expansion carbon layer has low thermal conductivity and can keep the flame zone away from the surface of the PET matrix, isolate the heat transfer phenomenon, reduce the temperature of the surface of the PET matrix, and inhibit the thermal decomposition process of PET, thereby stopping the provision of combustible gases to the flame zone, so that the flame retardancy is improved.

However, as with more DBDPE content, the quality of the carbon layer deteriorated due to a large amount of gas generated during the decomposition of DBDPE. For PET/2ZDP/10DBDPE and PET/12DBDPE, there was almost no residual char.

In order to further analyze the flame-retardant mechanism produced by compounding ZDP and DBDPE in different ratios, the micro-morphologies of the char layers of three groups of samples, namely PET/9ZDP/3DBDPE, PET/7ZDP/5DBDPE, and PET/4ZDP/8DBDPE, were observed, and these are shown in Figure 6 and Figure 7. The surfaces of the PET/9ZDP/3DBDPE and PET/7ZDP/5DBDPE samples maintained good densification and continuity with fewer holes. However, the surface quality of the carbon layer of PET/4ZDP/8DBDPE samples deteriorated significantly, with an increase in the number of holes. This is similar to the results obtained in Figure 5, indicating that the decrease in ZDP content reduced the quality of the char layer produced by PET combustion. As for the cross-section of the char layer, it can be seen that there are many holes in the char layers of the three samples, which show apparent characteristics of a porous char layer. Meanwhile, there are also a large number of hollow structures in the cross-sections of PET/9ZDP/3DBDPE and PET/7ZDP/5DBDPE samples, which significantly increases the height of the char layer, which is favorable to keep the PET matrix away from the flame zone and to inhibit the heat and mass transfer phenomena during the combustion process. On the other hand, the hollow structures of the PET/4ZDP/8DBDPE samples all collapsed, which led to a decrease in the thickness of the char layer, which was caused by the decrease in the ZDP content, resulting in the poor stability and mechanical strength of the residual char layer.

In summary, the char layer structure influences the flame-retardant properties of PET/ZDP/DBDPE. The flame-retardant properties of the blends increase with the thickness of the char layer, which suggests that the compounding of ZDP and DBDPE can produce an intumescent flame-retardant mechanism. ZDP/DBDPE, in appropriate ratios, can make the PET matrix obtain an excellent intumescent char layer structure during combustion, thus obtaining the best synergistic effect for the flame retardation of PET.

#### 3.1.4. Thermogravimetric Analysis

The thermal weight loss behavior of PET/ZDP/DBDPE samples was tested under air and nitrogen atmospheres, respectively, to analyze the effects of ZDP and DBDPE compounding on the thermal decomposition of PET.

Figure 8 shows the thermal weight loss curves of PET/ZDP/DBDPE blends at a total addition of 12 wt% ZDP/DBDPE under the N_2_ atmosphere, and the detailed data are listed in Table 4. The initial thermal decomposition temperature T_5%_ of PET was 394.8 °C, and the thermal decomposition temperatures were reduced to 387.9 °C and 378.1 °C with the addition of 12 wt% ZDP and DBDPE, respectively, while the PET/12DBDPE sample was reduced by 16.7 °C compared to the T_5%_ of PET, which indicated a significant reduction in the initial thermal stability of PET; this was related to the flame-retardant mechanism of ZDP and DBDPE. The decomposition temperature of ZDP is slightly lower than that of PET; in addition, H_2_PO_4_, anhydride, and other phosphoric acid substances are produced during ZDP thermal decomposition at high temperatures. They promote the dehydration of PET and result in a reduction in the mixture’s thermal stability [47]. DBDPE is a brominated flame retardant, mainly through the thermal decomposition of Br· and HBr radicals generated in the flame zone to capture the H· and OH· radicals to block the combustion chain reaction. Generally, the worse the thermal stability of halogens, the better their flame-retardant properties (under the conditions of meeting the polymer matrix processing temperature), which is the reason why the flame-retardant efficiency of the brominated flame retardant is higher than that of the chlorinated flame retardant [48]. Therefore, the thermal decomposition of DBDPE leads to a significant decrease in the thermal stability of PET/12DBDPE samples.

In the ZDP/DBDPE synergic flame-retardant samples, T_5%_ gradually decreased with the increase in DBDPE addition due to the lower thermal stability of DBDPE. Also, T_10%_ and T_max_ show the same trend as T_5%_, indicating that the thermal decomposition temperature zones of PET/ZDP/DBDPE blends move towards the low temperature zone with the increase in DBDPE content. In addition, the mass residual rate C_800,cal_ of PET/ZDP/DBDPE blends at 800 °C under theoretical conditions was calculated according to Equation (4), and the results are listed in Table 4. Compared with the theoretical mass residual rate, the actual mass residual rate C_800_ at 800 °C was higher, indicating that the compounding of ZDP and DBDPE contributes to the char formation of PET. From the mass residual rate C_800_ at 800 °C, it can be found that the mass residual rate of PET/ZDP/DBDPE blends at high temperatures showed a tendency of increasing and then decreasing with the increase in DBDPE content. The C_800_ of the samples with 12 wt% ZDP or DBDPE added are 12 wt% and 13.6 wt%, respectively, which are almost unchanged or even slightly decreased compared with the mass residual rate of pure PET, but the C_800_ of the PET/11ZDP/1DBDPE sample is obviously increased to 14.5 wt%; at the PET/7ZDP/5DBDPE sample, the C_800_ has the maximum value of 18.2 wt%, which is increased by 6.2 wt% compared to the PET/12ZDP sample; when the DBDPE content is higher than that of ZDP, the mass residual rate C_800_ at 800 °C starts to decrease. This suggests that the compounding of DBDPE and ZDP also promotes the char formation of PET at high temperatures and has the best synergistic effect at the appropriate ratio, while higher char formation will also produce better results for stabilizing the expanded char layer.
(4)$$C_{800/Cal}=C_{800/12Z}\;*\;\frac{x}{12}+C_{800/12D}\;*\;y/12$$
where *x* is the mass fraction of ZDP in 12 parts of flame retardant, and *y* is the mass fraction of DBDPE.

In addition, the compounding of ZDP and DBDPE also had an effect on the maximum thermal decomposition rate during the thermal decomposition of PET, and the maximum thermal decomposition rates (DTG peak) were reduced from 12.3 wt%/min for PET to 10.5 wt%/min and 9.9 wt%/min, respectively. With the addition of ZDP or DBDPE alone, it can be found that DBDPE was more effective in reducing the maximum thermal decomposition rate compared to ZDP, so in PET/ZDP/DBDPE blends, the DTG peak decreases with the increase in DBDPE content.

The thermal weight loss under the nitrogen atmosphere can simulate the situation in the region of thermal decomposition of the polyester when the sample is burned. The thermal weight loss curve under the air atmosphere can reflect the thermal decomposition of the part of the sample that is in contact with air when it is burned and the thermal stabilization of the char layer [49]. Figure 9 shows the thermal weight loss curves of PET/ZDP/DBDPE blends under the air environment, and the detailed data are listed in Table 5.

The carbon layer of PET produced in an air environment is oxidized at high temperatures, so the heat loss curve shows two stages. The first stage is the thermo-oxidative decomposition of PET/ZDP/DBDPE blends, as in the N_2_ environment. Due to the lower thermal stability of DBDPE compared to PET and ZDP, T_5%_, T_10%_, and T_max1_ all decrease with the increase in the DBDPE content, and residual char is formed in the interval of 450 °C–500 °C. The second stage occurs in the 500 °C–600 °C interval, which is dominated by the thermal oxidative decomposition of residual char, and the representative data of the peak temperature of the second stage of the thermal decomposition rate, T_max2_, can characterize the thermal stability of the char layer. The T_max2_ of PET was 535 °C. After adding ZDP, the char layer’s thermal oxidation slowed down, and the corresponding decrease in the weight loss curve became smooth, while the peak temperature was significantly increased to 579.6 °C. The addition of 12 wt% DBDPE reduced the T_max2_ of PET/12DBDPE to 518.7 °C, indicating a significant reduction in the thermal stability of the char layer. For PET/ZDP/DBDPE blends, T_max2_ increased along with the ZDP content, i.e., as the thermal stability of the char layer improved. This is because phosphate groups and other P=O structures can bind to the oxidation-prone sites on the surface of the char layer, deactivating the oxidatively active centers and, thus, increasing the thermal stability of the char layer [50,51]. Therefore, the presence of ZDP can significantly improve the stability of the expanded char layer during combustion, leading to a better structure at a higher ZDP content.

### 3.2. Foaming Performance Analysis

#### 3.2.1. Rheological Property

Melt viscoelasticity is an essential property of polymers and determines the foamability of PET. Industrially produced PET is a linear molecular chain with low melt viscoelasticity and poor strain hardening during stretching, so its foaming performance is weak. We chose to add 0.6 wt% of chain extender to PET/ZDP/DBDPE to increase the viscoelasticity by forming the branching chain on PET macromolecule. The chain extender (PMDA), PET, ZDP, and DBDPE were co-mixed in a torque rheometer, and the rotor torque was recorded with time. Figure 10 shows the torque curves for different ratios of ZDP/DBDPE. The time required for the chain extension reaction of PET is around 8 min. This time is drastically reduced to around 5 min after adding ZDP, which indicates that ZDP can effectively accelerate PET’s chain extension reaction. ZDP is a low-molecular-weight substance which acts as plasticizer and enhances the mobility of PET molecules, thereby accelerating the chain extension reaction. However, the final torque of the PET/ZDP/DBDPE mixture is slightly lower than that of the PET.

At 1 min, PET melted and plasticized under high temperature and shear stresses, and the torque decreased rapidly. The slight decrease in torque after melting is attributed to shear heat and the thermal degradation of PET. Around 2 min, the torque initially stabilized, and the difference in torque between PET/ZDP/DBDPE blends with different ratios of ZDP/DBDPE was relatively small, indicating that the effect of different ratios of ZDP/DBDPE on the plasticizing effect of PET at the processing temperature does not differ much. The torque starts to increase after 150 s due to the branching of PET under the action of the chain extender, and it reaches its maximum value at around 5–5.5 min. Figure 10 and Figure 11 show that the higher the content of ZDP, the earlier the torque starts to increase and the earlier the maximum torque value is reached, which indicates that DBDPE inhibits the reaction rate of PET chain extension. In this experiment, DBDPE is a solid particle at the processing temperature, because it is below its melting point (357 °C). As such during the mixing process, DBDPE hinders the movement of molecular chains and reduces the effect of the chain expansion reaction. Meanwhile, comparing the maximum torque values of PET/ZDP/DBDPE mixtures, it can be found that with the decrease in ZDP content, the torque shows a tendency to increase and then decrease, and there is a maximum value of 6.3 N·m in the case of the PET/7ZDP/5DBDPE samples. When the DBDPE content is higher than the ZDP content, the maximum torque value decreases significantly, and the maximum torque values of PET/4ZDP/8DBDPE and PET/2ZDP/10DBDPE are reduced to 4.2 N·m and 4 N·m, respectively.

In order to analyze the reason for the change in torque of PET/ZDP/DBDPE mixtures during chain extension, the rheological properties of the mixtures were tested on a rotational rheometer. The results are shown in Figure 12. In Figure 12a, the complex viscosity of all the samples decreases with the increase in ω, which exhibits a shear-thinning behavior, indicating that PET/ZDP/DBDPE mixtures are typical pseudoplastic fluids. However, it can be found that in the low-frequency region, the trend of the complex viscosity of PET changes more slowly, and there is a specific Newtonian plateau region. After the addition of ZDP, the complex viscosity of PET/ZDP/DBDPE blends, however, shows an upward behavior in the low-frequency region, which is attributed to the fact that ZDP promotes the solid state polycondensation of PET at high temperatures, leading to the polymerization of PET molecular chains at low shear under N_2_ blowing conditions, which results in a drastic increase in the complex viscosity phenomenon. This also leads to different trends in the loss angle tangent, storage modulus, and loss modulus curves in the low-frequency region for pure PET and PET/ZDP/DBDPE blends.

In the same trend as for the torque, the complex viscosity, storage modulus, and loss angle tangent of PET/ZDP/DBDPE blends in Figure 12a,c,d all show a tendency to increase and then decrease with decreasing ZDP content, with a maximum for PET/7ZDP/5DBDPE. In order to minimize the effect of solid-state polycondensation in the low-frequency region on the complex viscosity, the complex viscosity at ω = 1.35 rad/s was selected for comparison, which can be seen in Figure 13. The complex viscosity of the PET/12ZDP sample with the addition of 12 wt% ZDP was 1248 Pa·s, and the value gradually increased with the decrease in ZDP content, reaching a maximum value at the PET/7ZDP/5DBDPE sample of 1788 Pa·s; after that, when the DBDPE content was higher than the ZDP content, the complex viscosity decreased drastically to 1089 Pa·s and 960 Pa·s for the PET/4ZDP/8DBDPE and PET/2ZDP/10DBDPE samples, respectively, while the loss angle tangent shows the opposite trend; with the decrease in the ZDP content, the tanδ decreases first and then increases. Meanwhile, due to the solid-state condensation phenomenon during the rheological test, the PET/ZDP/DBDPE blends showed a peak near ω = 1 rad/s, while the tanδ of the PET samples decreased continuously with the increase in ω.

It can be seen that both ZDP and DBDPE reduce the melt viscoelasticity of PET, and a comparison of the rheological data of the PET/11ZDP/1DBDPE and PET/2ZDP/10DBDPE samples shows that DBDPE has a more significant effect on PET. The compatibility between PET, ZDP, and DBDPE can explain this. Table 6 shows the contact angles and calculated surface free energies of PET, ZDP, and DBDPE. The surface free energies of PET, ZDP, and DBDPE are 45 mN/m, 46 mN/m, and 72 mN/m, respectively. The surface free energies of PET and ZDP are closer to each other and much lower than that of DBDPE, which indicates that PET and ZDP are more compatible with each other than DBDPE.

Figure 14 shows the cross-section SEM image of PET/ZDP/DBDPE blends with different ZDP/DBDPE ratios. Due to the significant differences in the surface free energies of the ZDP and DBDPE phases, they are dispersed in a multinuclear structure in the PET matrix. In the PET/11ZDP/1DBDPE figure, ZDP can be uniformly distributed in the PET matrix with a particle size of around 3 µm, but it is still partially agglomerated. Moreover, in the PET/2ZDP/10DBDPE sample, DBDPE is more agglomerated, with a larger particle size of around 5–10 µm, which indicates that DBDPE has a more significant impact on the melt viscoelasticity of PET. However, the cross-sections of the PET/9ZDP/3DBDPE and PET/7ZDP/5DBDPE samples do not show large-size DBDPE agglomerates. At the same time, the agglomeration phenomenon of ZDP is significantly reduced, and both have better dispersion in the PET matrix. This is because when the content of ZDP or DBDPE is high, the two are prone to agglomeration, resulting in poorer dispersibility. With the addition of ZDP/DBDPE ratio close to 1, the relative content of ZDP and DBDPE is smaller. The two are dispersed in the PET matrix with a multinuclear structure and will not be attracted to each other. The two components will not be agglomerated, so the two are dispersed in the best way, which also minimizes the damage to the viscoelasticity of the PET melt.

#### 3.2.2. Foaming Properties

Batch foaming experiments were carried out to test the foamability of PET/ZDP/DBDPE blends, and Figure 15 demonstrates the foaming ratio of each sample at different temperatures. The foaming temperature window of PET is vast, and the foaming ratio is more than 10 when the foaming temperature is higher than 220 °C, while the maximum foaming rate is 41 times at 235 °C. Even when the temperature reaches 250 °C, the foaming ratio remains 26 times The foaming ratio of the PET/ZDP/DBDPE mixtures has the same relationship with the temperature as demonstrated by the PET. When the temperature increases, the mixture’s foaming ratio increases first and then decreases, and there is a maximum value when the foaming temperature is about 235 °C. This is because the temperature influences both melt viscoelasticity and CO_2_ solubility parameters during foaming: both the melt viscoelasticity and CO_2_ solubility of PET/ZDP/DBDPE blends are negatively correlated with temperature. During the bubble growth stage, the cell wall is subjected to tensile forces in both directions. If the melt viscoelasticity is too low, the cell wall will break during the stretching process, causing bubble rupture or merger. If the melt viscoelasticity is too high, it will limit the growth of the bubble, so an appropriate melt viscoelasticity is the key to producing good PET foam products. In the range of 220–235 °C, melt viscoelasticity plays a significant role in the foaming process. The inhibition of cell growth decreases with melt viscoelasticity, and the size of the bubble increases. The solubility of CO_2_ plays a significant role in the foaming process at temperatures ranging from 235 to 250 °C. As the solubility of CO_2_ decreases, there is a reduction in cell nucleation, leading to decreased cell density in the foaming material. At the same time, the strength of the melt is low at high temperatures, resulting in more and more bubbles breaking and merging. The cell diameter and a foaming ratio are listed in Table 7. 

From Figure 15 and Figure 16, it can be seen that the foaming performance of PET was reduced by the addition of ZDP/DBDPE; especially at a high DBDPE content, the foaming ratio at all temperatures was reduced dramatically, and the maximum foaming ratio of PET/4ZDP/8DBDPE and PET/2ZDP/10DBDPE samples were only 12 and 10 times, respectively. With the decrease in ZDP content, PET/ZDP/DBDPE blend foaming performance tended to trend first up and then down. The suitable foaming temperature window of PET/11ZDP/1DBDPE samples was 225 °C–240 °C, and the maximum foaming multiplicity was 34 times, while that of the PET/9ZDP/3DBDPE samples was 230 °C–245 °C and the maximum foaming ratio was 43 times greater than usual. The PET/7ZDP/5DBDPE samples of the appropriate foaming temperature window were 225 °C–245 °C, and the maximum foaming ratio was 39 times greater than usual. When DBDPE continued to increase, the foaming ratio decreased rapidly. The mixture obtained the best foaming performance at PET/7ZDP/5DBDPE. The reason is that the melt viscoelasticity of the blends is reduced compared with PET after adding ZDP/DBDPE, which reduces foaming performance at high temperatures when the bubble breaks at the bubble growth stage and fails to maintain the cell morphology. The foaming performance changes with the additional ratio of ZDP and DBDPE, also due to the melt viscoelasticity of the PET/ZDP/DBDPE mixture increasing and then decreasing with the decrease in ZDP content. The PET/7ZDP/5DBDPE sample has the highest melt viscoelasticity and the best foaming performance. When the content of DBDPE is higher than that of ZDP, the melt viscoelasticity of the mixture decreases significantly, and the foaming performance also decreases. In Table 7, it can be seen that the diameter of the cell decreases with decreasing ZDP content while the density of the cell increases, which indicates that the cell structure is also affected by the change in melt viscoelasticity. Figure 17 shows the corresponding SEM images of the cell structures.

### 3.3. Flame-Retardant Properties of Foamed Samples

Flame-retardant masterbatches were formulated by blending ZDP/DBDPE in ratios of 2:1 and 3:1 with the PET matrix. Flame-retardant PET foam samples were prepared using an extrusion foaming line, as shown in Figure 18. The limiting oxygen index and vertical flammability level of PET foam samples are listed in Table 8.

As seen from Table 8, the foamed samples with a ratio of 3 for ZDP and DBDPE have a higher limiting oxygen index than the samples with a ratio of 2 at the same additive amount. This may be because the mass of the sample is reduced after foaming, and the carbon layer generated is reduced. However, the area that needs to be protected remains the same, so more phosphorus flame retardants are needed to promote the formation of the carbon layer and to improve the flame-retardant effect.

Figure 19 shows the effect of sample density on the limiting oxygen index of the samples after foaming. The limiting oxygen index of all foamed samples decreased somewhat as the density of the samples decreased. A lower density means that the samples have a larger contact area with the flame and air during combustion and are more prone to pyrolysis, reducing flame retardancy.

## 4. Conclusions

To improve the flame-retardant properties of PET foams, a ZDP/DPDPE synergistic system was chosen to modify PET and investigate its impact on the flame-retardant and foaming properties of PET. The main conclusions are as follows.

Adding ZDP and DBDPE alone increased the oxygen index of PET to 29.8% and 28.7%, respectively, but neither improved the dripping phenomenon, resulting in a UL-94 rating of V-2. The compounding of ZDP and DBDPE is a synergistic mechanism for expanding the carbon layer. When the ZDP/DBDPE = 9/3 and 7/5, the residual carbon layer reaches 3–4 cm at the highest level; a porous char layer with a considerable thickness can isolate the heat transfer between the PET matrix and the flame zone and inhibit the thermal decomposition of the PET matrix. The LOI values of the PET/9ZDP/3DBDPE and PET/7ZDP/5DBDPE samples with the most considerable char layer thicknesses reached 32.7% and 33.6%, respectively, and the vertical combustion test also reached the V-0 level.

ZDP and DBDPE have a significant effect on the foaming properties of the PET matrix. Because ZDP is more compatible with PET than DBDPE, increasing the proportion of ZDP can improve the viscoelasticity of the melt at a fixed total addition amount, thus enhancing the foaming performance. In addition, the dispersion of ZDP and DBDPE in PET is critical. An appropriate ratio (PET/7ZDP/5DBDPE) optimizes the dispersibility and reduces the adverse effect on the melt viscoelasticity. This optimization not only enlarges the foaming temperature window but also significantly improves the foaming efficiency, reaching a maximum foaming ratio of 39 times.

The flame retardancy of foamed samples is a function of foam density. A decrease in the density of the foamed samples correlates with a reduction in the LOI, even when an equivalent quantity of flame retardant is incorporated. When the density of the foam is significantly reduced, the samples may exhibit dripping of molten material during combustion, resulting in a vertical burning classification of V-2. Additionally, the ratio of ZDP to DBDPE significantly influences the flammability characteristics of the foam.

## Figures and Tables

**Figure 1 polymers-16-01690-f001:**
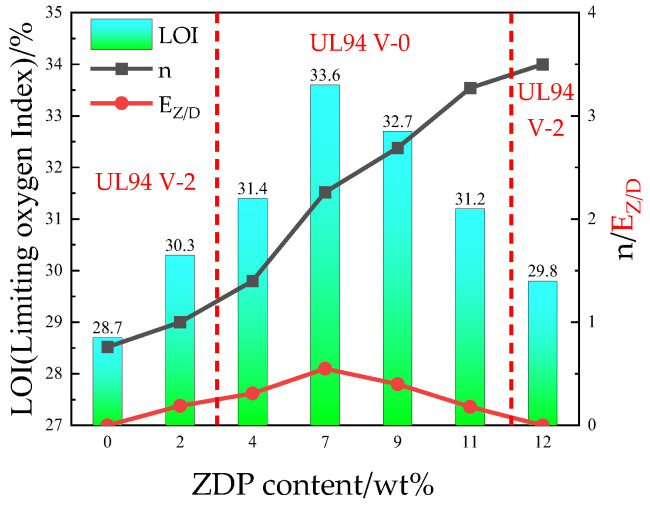
Effects of ZDP content on the flame retardancy of PET/ZDP/DBDPE at 12 wt% ZDP/DBDPE.

**Figure 2 polymers-16-01690-f002:**
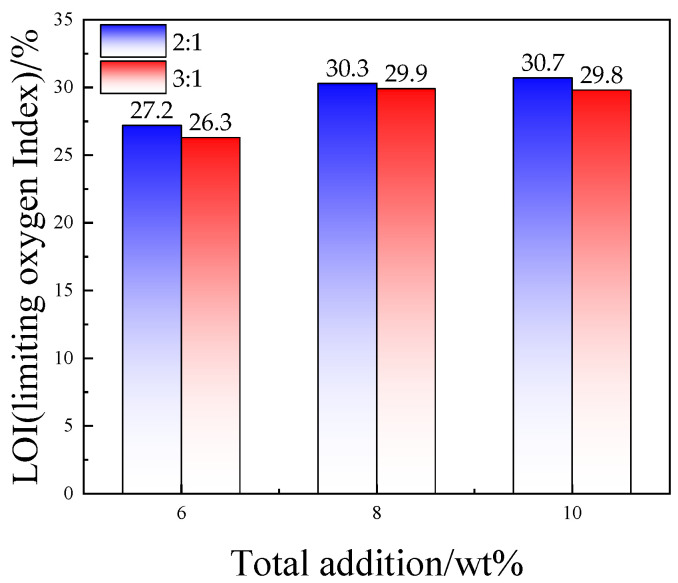
Effect of total flame retardancy on limiting oxygen index of PET/ZDP/DBDPE composites when the ratios of ZDP/DBDPE are 2 and 3, respectively.

**Figure 3 polymers-16-01690-f003:**
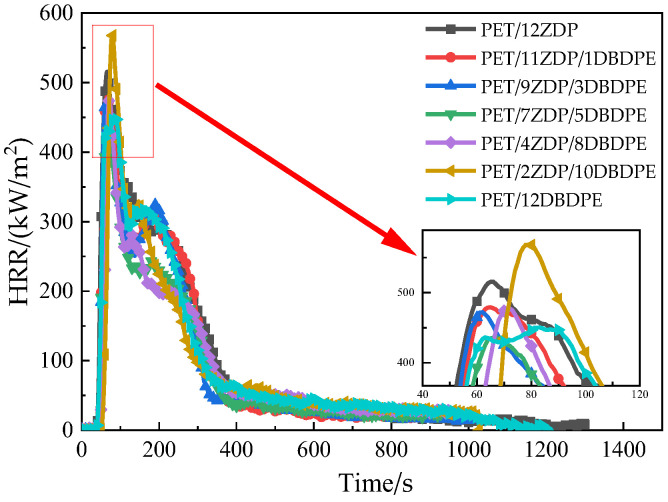
Heat release rate.

**Figure 4 polymers-16-01690-f004:**
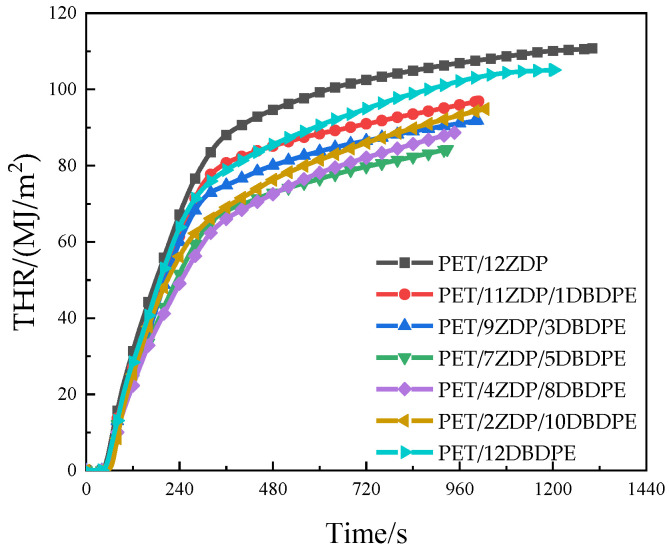
Total heat release.

**Figure 5 polymers-16-01690-f005:**
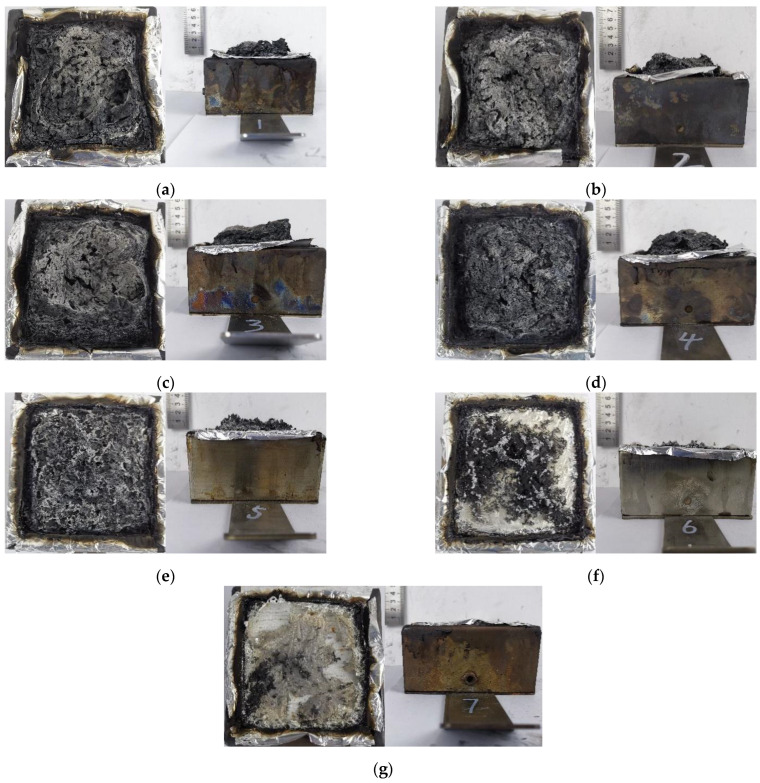
Carbon layer after cone calorimeter test. (**a**) PET/12ZDP; (**b**) PET/11ZDP/1DBDPE; (**c**) PET/9ZDP/3DBDPE; (**d**) PET/7ZDP/5DBDPE; (**e**) PET/4ZDP/8DBDPE; (**f**) PET/2ZDP/10DBDPE; (**g**) PET/12DBDPE.

**Figure 6 polymers-16-01690-f006:**
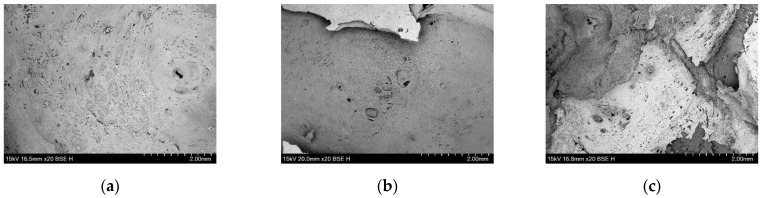
Micromorphology of carbon layer surfaces. (**a**) PET9ZDP/3DBDPE; (**b**) PET/7ZDP/5DBDPE; (**c**) PET/4ZDP/8DBDPE.

**Figure 7 polymers-16-01690-f007:**
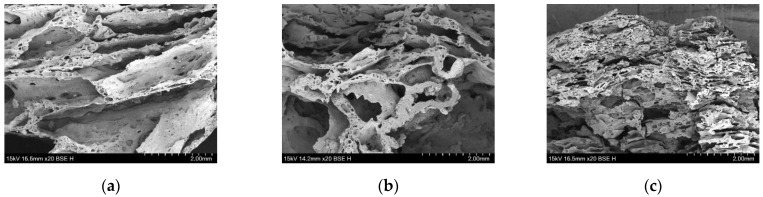
Micromorphology of carbon layer cross-sections. (**a**) PET9ZDP/3DBDPE; (**b**) PET/7ZDP/5DBDPE; (**c**) PET/4ZDP/8DBDPE.

**Figure 8 polymers-16-01690-f008:**
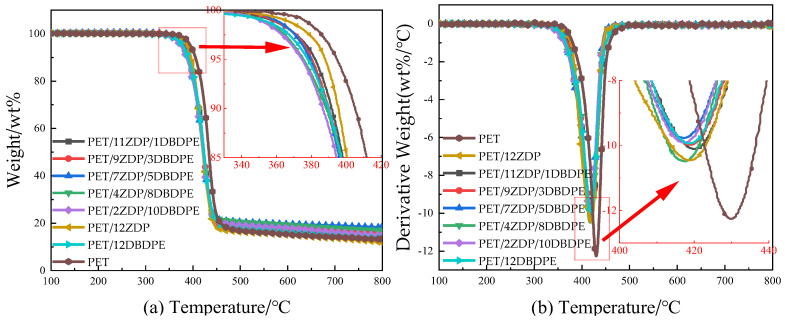
Thermogravimetric curves of PET/ZDP/DBDPE composites in an N_2_ environment. (**a**) Thermogravimetric curve (TG); (**b**) differential thermogravimetric curve (DTG).

**Figure 9 polymers-16-01690-f009:**
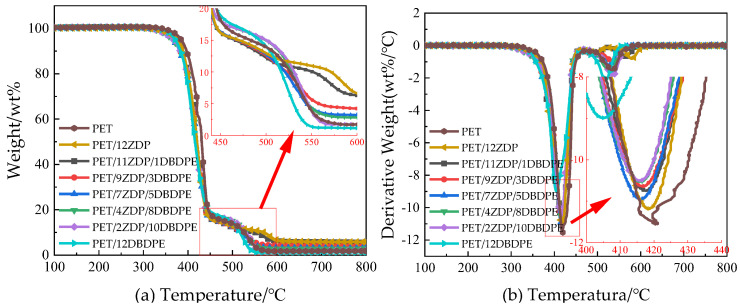
Thermogravimetric curve of PET/ZDP/DBDPE composites in an air atmosphere. (**a**) Thermogravimetric curve (TG); (**b**) differential thermogravimetric curve (DTG).

**Figure 10 polymers-16-01690-f010:**
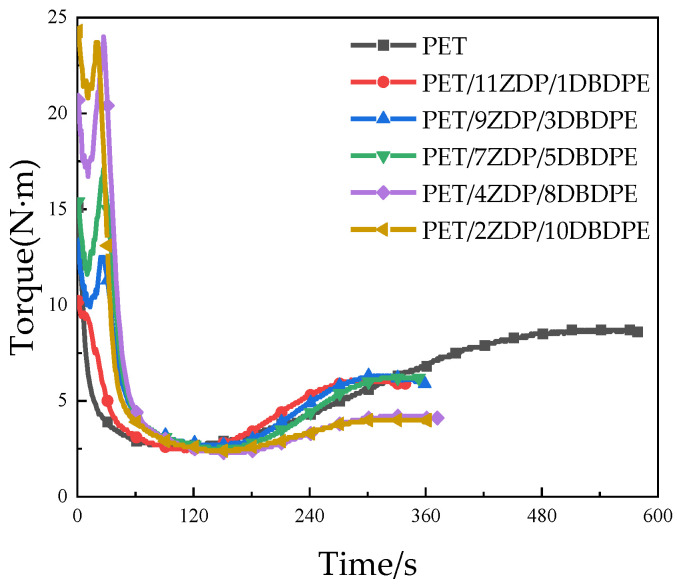
Torque curves vs. time of PET/ZDP/DBDPE composites during melt blending.

**Figure 11 polymers-16-01690-f011:**
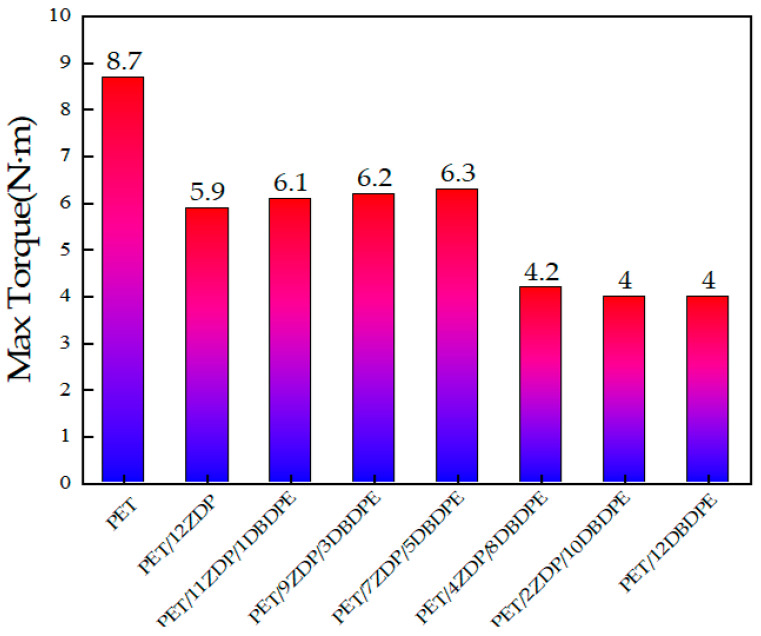
Maximum torque of PET/ZDP/DBDPE composites during melt blending.

**Figure 12 polymers-16-01690-f012:**
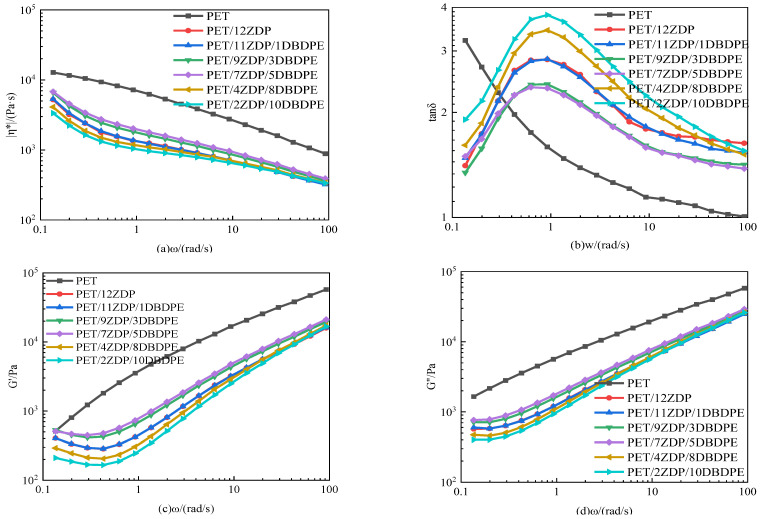
Rheological properties of PET/ZDP/DBDPE composites. (**a**) Complex viscosity |η*|; (**b**) loss tangent (tanδ); (**c**) storage modulus (G’); (**d**) loss modulus (G”).

**Figure 13 polymers-16-01690-f013:**
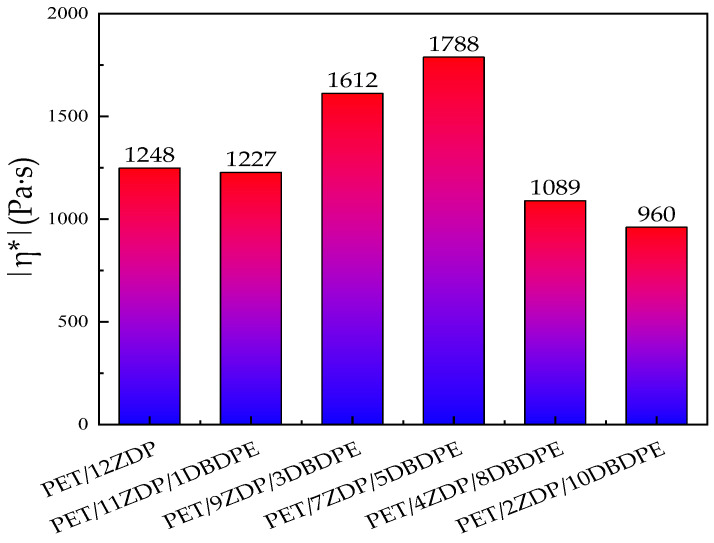
Complex viscosity of PET/ZDP/DBDPE composites with *ω* = 1.35 rad/s.

**Figure 14 polymers-16-01690-f014:**
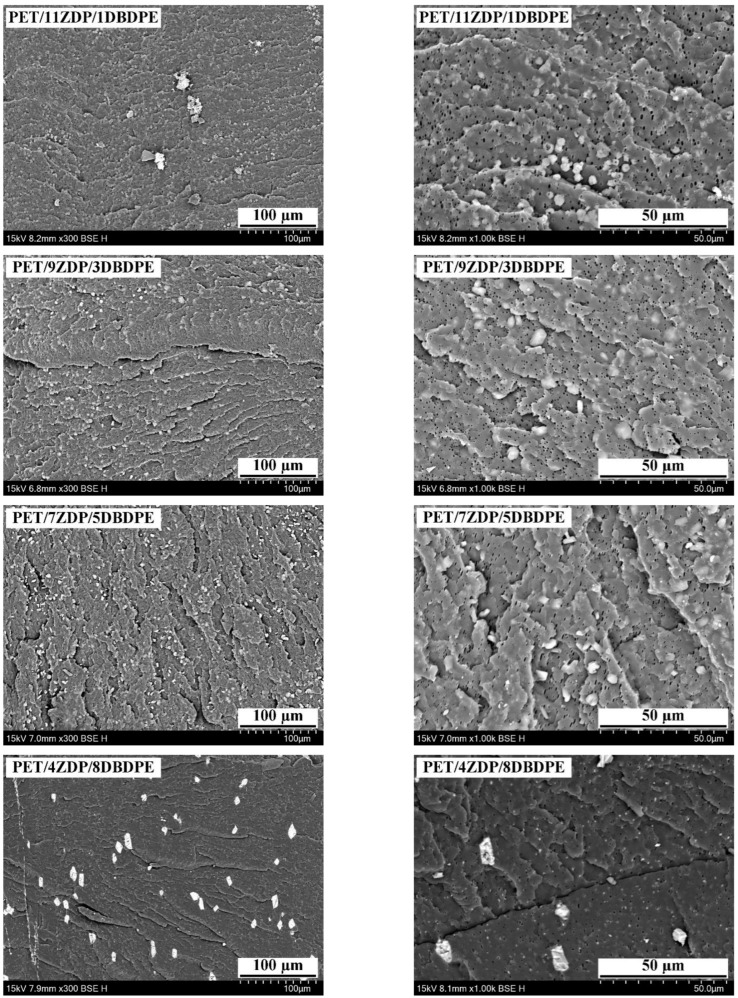
SEM of PET/ZDP/DBDPE composites Cross section.

**Figure 15 polymers-16-01690-f015:**
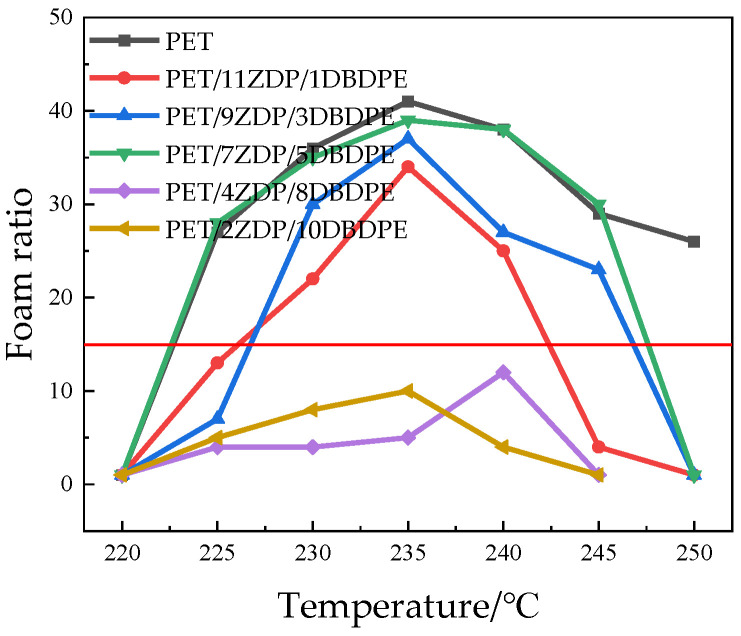
Foaming ratios of PET/ZDP/DBDPE composites at different temperatures (a foaming ratio greater than 10 indicates better foaming performance).

**Figure 16 polymers-16-01690-f016:**
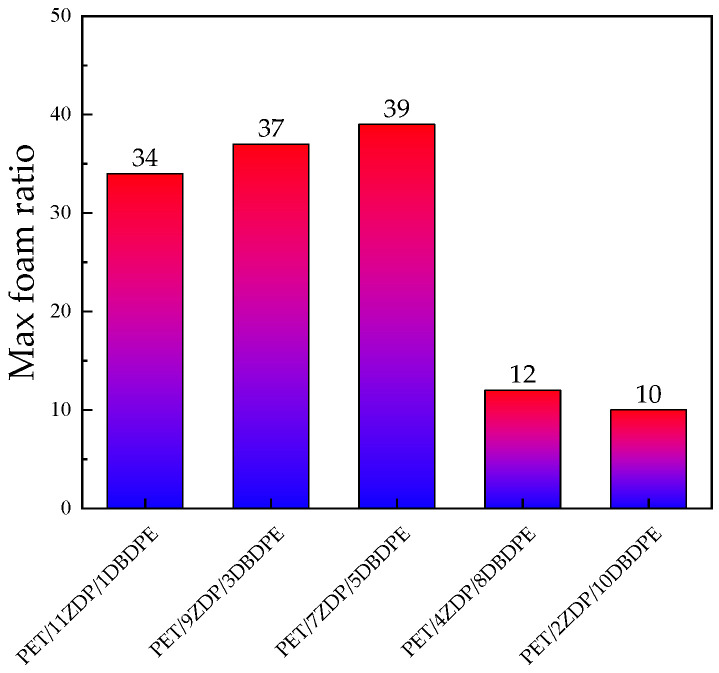
The maximum foaming ratios of PET/ZDP/DBDPE composites.

**Figure 17 polymers-16-01690-f017:**
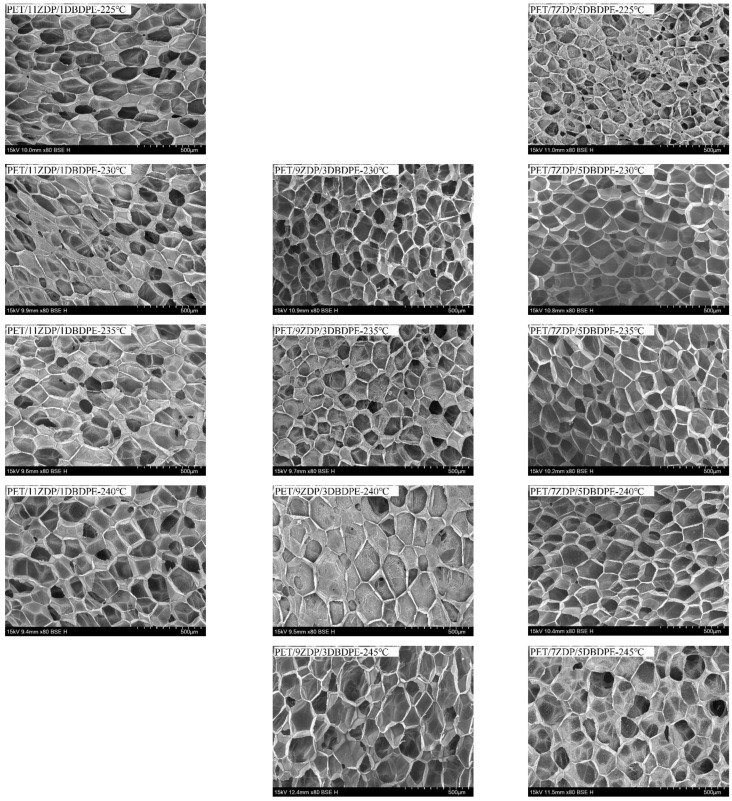
Cell structures of PET/ZDP/DBDPE composites after foaming.

**Figure 18 polymers-16-01690-f018:**
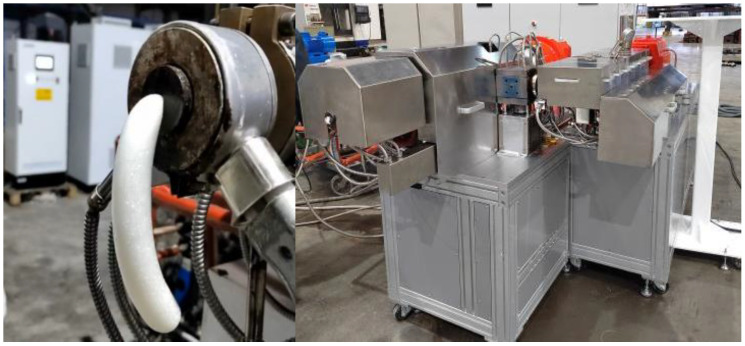
Extrusion foaming line.

**Figure 19 polymers-16-01690-f019:**
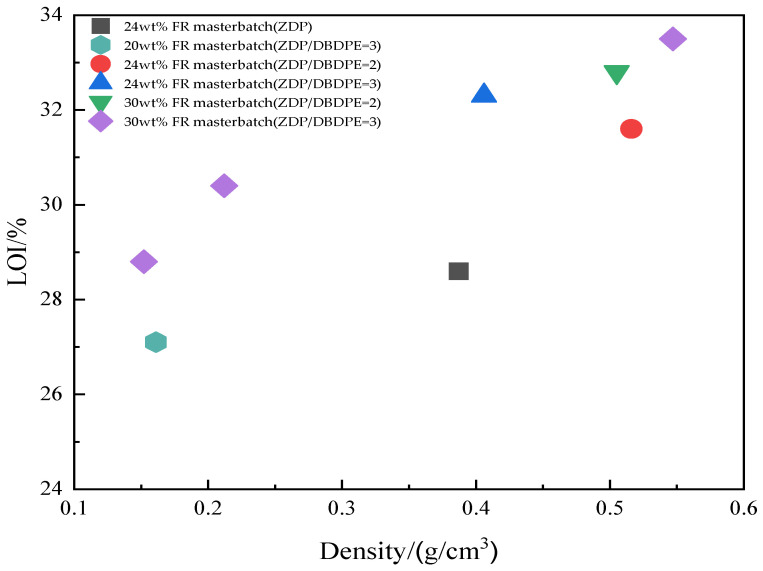
Effect of density on the limiting oxygen index of foaming samples.

**Table 1 polymers-16-01690-t001:** Experiment formula (wt%).

Composition	PET	ZDP	DBDPE	PMDA
PET/12ZDP	88	12	0	0.6
PET/11ZDP/1DBDPE	88	11	1	0.6
PET/9ZDP/3DBDPE	88	9	3	0.6
PET/7ZDP/5DBDPE	88	7	5	0.6
PET/4ZDP/8DBDPE	88	4	8	0.6
PET/2ZDP/10DBDPE	88	2	10	0.6
PET/12DBDPE	88	0	12	0.6

**Table 2 polymers-16-01690-t002:** Flame retardancy of PET/ZDP/DBDPE composites.

Samples	t_1_ + t_2_/s	Dripping	UL-94	LOI/%	*Cp* ^1^/%	*C_Br_* ^2^/%	*n*	*E_Z/D_*
PET	≥30	Yes	NR	21.6	0	0	--	--
PET/12ZDP	≤10	Yes	V-2	29.8	2.4	0	3.4	0
PET/11ZDP/1DBDPE	6.58	No	V-0	31.2	2.2	0.8	3.2	0.18
PET/9ZDP/3DBDPE	3.68	No	V-0	32.7	1.8	2.4	2.6	0.40
PET/7ZDP/5DBDPE	3.42	No	V-0	33.6	1.4	4.0	2.2	0.55
PET/4ZDP/8DBDPE	2.56	No	V-0	31.4	0.8	6.4	1.4	0.31
PET/2ZDP/10DBDPE	2.62	Yes	V-2	30.3	0.4	8.0	1.0	0.19
PET/12DBDPE	≤10	Yes	V-2	28.7	0	9.6	0.7	0

^1^ Phosphorus content; ^2^ bromine content.

**Table 3 polymers-16-01690-t003:** Analysis data of cone calorimeter test.

Sample	TTI ^1^/s	PkHRR ^2^/(kW/m^2^)	THR ^3^/(MJ/m^2^)	EHC ^3^/(MJ/kg)	TSP ^3^/m^2^	CO Yield ^3^/(kg/kg)
PET/12ZDP	40	516	111	27	22	0.35
PET/11ZDP/1DBDPE	44	479	97	22	20	0.38
PET/9ZDP/3DBDPE	44	472	92	26	19	0.49
PET/7ZDP/5DBDPE	42	437	84	22	20	0.51
PET/4ZDP/8DBDPE	45	477	89	23	20	0.51
PET/2ZDP/10DBDPE	53	568	95	24	20	0.46
PET/12DBDPE	40	450	105	23	24	0.37

^1^ Time to ignition; ^2^ peak of heat release rate; ^3^ average of the last 6 min.

**Table 4 polymers-16-01690-t004:** Tg analysis data of PET/ZDP/DBDPE composites in an N_2_ environment.

Sample	T_5%_ ^1^/°C	T_10%_ ^2^/°C	T_max_ ^3^/°C	R_max/_(wt%/min)	C_600_ ^4^/wt%	C_800_ ^5^/wt%	C_800,cal_ ^6^/wt%
PET	394.8	405.1	430.8	12.3	15.2	13.4	--
PET/12ZDP	387.9	395.6	418.8	10.5	15.0	12	--
PET/11ZDP/1DBDPE	382.3	391.6	420	10.1	16.7	14.5	12.1
PET/9ZDP/3DBDPE	380.3	390.2	418.8	10.0	17.7	16.3	12.4
PET/7ZDP/5DBDPE	379.3	389.2	417.3	9.8	19.6	18.2	12.7
PET/4ZDP/8DBDPE	375.5	386.6	417.3	10.5	19.2	17.1	13.1
PET/2ZDP/10DBDPE	374.7	386.5	417.3	9.9	17.8	14.8	13.3
PET/12DBDPE	378.1	389.8	418.8	9.9	15.9	13.6	--

^1^ Temperature at which the sample loses 5 wt% of its weight; ^2^ temperature at which the sample loses 5 wt% of its weight; ^3^ temperature at which the maximum rate of thermal decomposition occurs; ^4^ actual mass residuals at 600 °C; ^5^ actual mass residuals at 800 °C; ^6^ calculated mass residuals at 800 °C.

**Table 5 polymers-16-01690-t005:** TG data of PET/ZDP/DBDPE composites in an air environment.

Sample	T_5%_/°C	T_10%_/°C	T_50%_ ^1^/°C	T_max1_ ^2^/°C	T_max2_ ^3^/°C	C_500_ ^4^/(wt%/min)	C_800_(wt%/min)
PET	384.7	397	430.8	421.1	535	13.9	1.9
PET/12ZDP	377.7	388.8	420.2	418.4	579.6	12.9	5.8
PET/11ZDP/1DBDPE	375.7	386.7	420.6	417.7	565.6	12.2	5.7
PET/9ZDP/3DBDPE	369.8	383.3	419.6	416.9	533.5	12.6	3.7
PET/7ZDP/5DBDPE	367.4	381.9	419.2	415.9	528.2	12.8	3.0
PET/4ZDP/8DBDPE	365.6	383.7	419.4	415.5	528.5	14.9	2.9
PET/2ZDP/10DBDPE	367.2	383.1	419.2	415.5	525.2	15.0	1.9
PET/12DBDPE	371.2	383.3	415.9	405.1	518.7	14.1	1.3

^1^ Temperature at which the sample loses 50 wt% of its weight; ^2^ temperature at which the first peak of the heat loss rate in the DTG curve is reached; ^3^ temperature at which the second peak of the heat loss rate in the DTG curve is reached; ^4^ actual mass residual rate at 500 °C.

**Table 6 polymers-16-01690-t006:** Contact angle test of PET, ZDP, and DBDPE.

Sample	Contact Angle ^1^/°	Contact Angle ^2^/°	Surface Free Energy/(mN/m)
PET	64.32	38.43	45
ZDP	57.378	48.66	46
DBDPE	129.55	12.04	72

^1^ Contact angle with water; ^2^ contact angle with diiodomethane.

**Table 7 polymers-16-01690-t007:** Cell density and cell diameter of PET/ZDP/DBDPE composites at different foaming temperatures.

Sample	Temperatura/°C	Cell Diameter/µm	Cell Density/(10^6^/cm^3^)
11Z/1D	225	118	5.7
	230	124	7.1
	235	152	8.0
	240	161	4.6
9Z/3D	230	117	14.1
	235	133	9.1
	240	159	4.8
	245	158	3.8
7Z/5D	225	94	12.8
	230	116	18.4
	235	122	17.2
	240	130	13.1
	245	164	5.9

**Table 8 polymers-16-01690-t008:** Flame retardancy of PET foams.

PET Foam Sample	FR Content (%)	Density/(g/cm^3^)	t_1_ + t_2_/s	Dripping	UL-94 Grade	LOI/%
24 wt% FR masterbatch (ZDP)	12	0.387	4.7	No	V-0	28.6
24 wt% FR masterbatch (ZDP:DBDPE = 2)	12	0.516	0	No	V-0	31.6
30 wt% FR masterbatch (ZDP:DBDPE = 2)	15	0.505	0	No	V-0	32.8
20 wt% FR masterbatch (ZDP:DBDPE = 3)	10	0.161	2.8	No	V-0	27.1
24 wt% FR masterbatch (ZDP:DBDPE = 3)	12	0.406	0.7	No	V-0	32.3
30 wt% FR masterbatch (ZDP:DBDPE = 3)	15	0.547	1.3	No	V-0	33.5
		0.212	0.5	No	V-0	30.4
		0.152	2.2	No	V-0	28.8

## Data Availability

The data used in this research can be found in the article.

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
