# Peer review of "Phosphorus/Bromine Synergism Improved the Flame Retardancy of Polyethylene Terephthalate Foams"

_polymers, 2024, doi:10.3390/polym16121690_

Round 1
Reviewer 1 Report
Comments and Suggestions for Authors
The authors report the synergistic use of ZDP/DPDPE in PET foams to produce a flame retardancy effect and improve foaming properties. In the study, it was found that the oxygen index increases; however, the dripping phenomenon is not optimized. Finally, ZDP increases foaming performance. Some points of the manuscript need to be improved.
Point 1. On line 132, check the phrase in title 2.2 on the word “retarded”.
Point 2. In section 2.3.6. It is suggested that the equipment used to carry out the analysis be indicated.
Point 3. Lines 494-495 mention that ZDP contributes to the chain extension reaction. It is recommended to justify how this material accelerates the reaction.
Point 4. In lines 511-512, reference is made to the effect of decreasing the chain reaction when using DBDPE. It is suggested to justify this reagent's chemical interaction with the polymer matrix to generate said effect.
Point 5. It is recommended that the format of Figures 8, 9, 12, and 15 be improved to make it more suitable for user interpretation.
Point 6. Lines 603-605 indicate that the viscoelasticity of the melt state is essential; however, the effect or interaction that this property generates in foam formation is not explained. It is recommended that this phenomenon be justified fully.
Point 7. Line 605 presents confusion in the structure of the sentence, and in the same way, it is suggested that the aforementioned phenomenon of bubble collapse be broadly discussed.
Author Response
Please see the file attached.

Reviewer 2 Report
Comments and Suggestions for Authors
Please see the file attached.

Author Response
Please see the file attached.

Round 2
Reviewer 1 Report
Comments and Suggestions for Authors
The authors made significant changes to the manuscript according to the observations and comments made. Therefore, its publication is recommended.
Reviewer 2 Report
Comments and Suggestions for Authors
The authors have addressed all my comments, and the manuscript seems to be suitable for publication.